# Machine Learning for Blockchain: Literature Review and Open Research Questions

**Luyao Zhang**[*]
Data Science Research Center and Social Science Division
Duke Kunshan University
8 Duke Ave., Kunshan, Suzhou, Jiangsu, China 215316
`lz183@duke.edu`

## Abstract

In this research, we explore the nexus between artificial intelligence (AI) and blockchain, two paramount forces steering the contemporary digital era. AI, replicating human cognitive functions, encompasses capabilities from visual discernment to complex decision-making, with significant applicability in sectors such as healthcare and finance. Its influence during the web2 epoch not only enhanced the prowess of user-oriented platforms but also prompted debates on centralization. Conversely, blockchain provides a foundational structure advocating for decentralized and transparent transactional archiving. Yet, the foundational principle of "code is law" in blockchain underscores an imperative need for the fluid adaptability that AI brings. Our analysis methodically navigates the corpus of literature on the fusion of blockchain with machine learning, emphasizing AI's potential to elevate blockchain's utility. Additionally, we chart prospective research trajectories, weaving together blockchain and machine learning in niche domains like causal machine learning, reinforcement mechanism design, and cooperative AI. These intersections aim to cultivate interdisciplinary pursuits in AI for Science, catering to a broad spectrum of stakeholders.

## 1 Introduction

In the rapidly evolving digital age, technologies like artificial intelligence (AI) and blockchain are at the forefront of transformative change (Salah et al., 2019; Dinh and Thai, 2018). Artificial Intelligence (AI), at its core, refers to the simulation of human intelligence in machines, enabling them to perform tasks that typically require human intellect such as visual perception, speech recognition, and decision-making. Its applications span various domains, from healthcare to finance, driving efficiency and innovation (Wang et al., 2023). Artificial Intelligence (AI) has been a cornerstone in the evolution of web2, serving as the driving force behind many of its innovations. Web2, often referred to as the "social web," represents the phase where the internet transitioned from static web pages (web1) to dynamic platforms that promote user-generated content, collaboration, and interactivity (Zhuravskaya et al., 2020). Platforms like social media, blogs, wikis, and video-sharing sites exemplify the web2 era. AI incubated within this framework, empowering these platforms with personalized content recommendations, targeted advertising, and enhanced user experiences. As AI algorithms processed vast amounts of user data, they facilitated the rapid growth and influence of web2 platforms. However, this symbiotic relationship between AI and web2 also sowed the seeds for the centralization concerns we face today, prompting discussions about the transition to a more decentralized web3 (Wan et al., 2023). Blockchain, often heralded as the decentralized artificial

---

[*]website: `https://scholars.duke.edu/person/luyao.zhang`

NeurIPS 2023 AI for Science Workshop.

intelligence and the foundational infrastructure of web3, promises to reshape industries by providing a secure, transparent, and decentralized way of recording transactions (Cao, 2022).

Blockchain, often celebrated for its groundbreaking transparency and decentralization, has introduced a new paradigm of secure and trustless transactions. Its decentralized ledger system ensures that every transaction is transparently recorded across multiple nodes, eliminating intermediaries and reducing centralized control risks. However, despite its revolutionary promise, blockchain confronts several challenges Gadekallu et al. (2022). A notable limitation is the term "smart contract," which, ironically, isn't truly "smart" (Zou et al., 2019). While these self-executing contracts with the terms of the agreement directly written into code lines can automate and streamline processes, they lack the adaptability and learning capabilities inherent in AI systems. The algorithms deployed on the blockchain are rigid and inflexible, operating strictly within their pre-defined parameters. This rigidity means that once deployed, these algorithms cannot learn, adapt, or evolve based on new data or changing conditions, unlike AI models. The absence of AI integration in blockchain systems, especially in the realm of smart contracts, hampers the technology's efficiency, scalability, and versatility. In the fast-changing world where what is optimal today can become suboptimal tomorrow, and contracts often grapple with unforeseen circumstances, the integration of AI with learning capacity becomes crucial. For blockchain to truly serve as the foundational infrastructure of web3, it must evolve and adapt dynamically, ensuring it remains relevant and effective in an ever-evolving digital landscape.

Why AI? Blockchain, with its foundational tenets of security and decentralization, adheres to the philosophy that "code is law" (De Filippi and Hassan, 2018). In essence, blockchain establishes decentralized trust through unwavering commitments to rules pre-ordained and deployed in open-source code. Direct human intervention would compromise this core principle. AI serves as a harmonious bridge in this context. Through machine learning, AI can endow blockchain with intelligence while still upholding transparency and dedication to learning rules rooted in collective agreements.

As illustrated in Figure 1, this paper offers an in-depth review of cutting-edge literature concerning the integration of blockchain with machine learning (ML). We emphasize the critical role of ML in enhancing the current infrastructure, applications, and cross-chain solutions of blockchain. A focal point of our discussion is blockchain's inherent security and decentralized nature, coupled with its growing need for sophisticated intelligence. Moreover, we venture into future research directions, intersecting blockchain and ML in domains such as causal machine learning, reinforcement mechanism design, and cooperative AI. Our objective is to foster interdisciplinary research in the

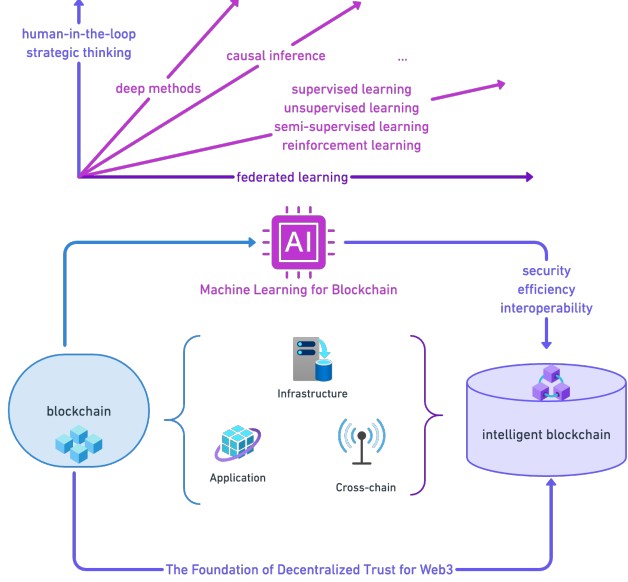

Figure 1: Machine Learning for Blockchain

realm of AI for Science with the potential to benefit a wide spectrum of stakeholders, from individual users to larger organizations.

## 2 Literature Review

Table 1 and Table 2 showcase a curated selection of literature that intersects machine learning and blockchain. These works are categorized based on three distinct facets:

1. **Infrastructure**: This facet pertains to the underlying foundational technologies and protocols that support blockchain networks, including consensus algorithms and the systems that ensure the secure and transparent recording of transactions. Studies by Qu et al. (2022); Chen et al. (2021); Mallouh et al. (2021); Jiang et al. (2016) explore the enhancement of blockchain consensus through the application of diverse machine learning methods. Conversely, research presented in Wang et al. (2021); Hou et al. (2019); Bar-Zur et al. (2022) investigates how reinforcement learning can amplify selfish mining strategies, thereby potentially compromising blockchain security at the infrastructure layer.

2. **Application**: This dimension focuses on the practical use cases and services built atop the blockchain infrastructure. It encompasses everything from financial services, such as cryptocurrencies, to decentralized applications (dApps) that operate on a blockchain network without centralized authority. Zhu et al. (2023); Li et al. (2022); Chen et al. (2021); Liu et al. (2020); Waheed et al. (2020); Salah et al. (2019); Tanwar et al. (2019); Nadeem et al. (2023) provides comprehensive surveys on the utilization of machine learning algorithms to enhance blockchain applications in domains such as the Internet of Things (IoT), 5G, metaverse, decentralized finance, cloud computing, and more. In a distinct study, Gai et al. (2023) leverages large language models to identify anomalous transactions in decentralized applications, aiming to bolster security alerts. Meanwhile, Liu and Zhang (2023) employs explainable machine-learning techniques for cryptocurrency valuations. Both Liu et al. (2022) and Zhang and Zhang (2023) utilize causal inference to analyze the impact of policy shifts in transaction fee mechanisms on the Ethereum blockchain.

3. **Cross-chain solutions**: This facet delves into the interoperability between different blockchain networks, enabling them to communicate, share, and exchange information seamlessly. Qu et al. (2022) delves into the advantages of federated learning (FL) for blockchain, highlighting its role in facilitating privacy-preserving cross-chain data exchanges and introducing an innovative energy-efficient consensus algorithm that integrates FL, thereby optimizing computational resources and significantly reducing energy consumption.

The tables further highlight the specific scientific methods employed in each piece of literature: *a) Theoretical*: This method involves abstract modeling and reasoning without direct empirical testing; *b) Empirical*: This approach relies on observation or experimentation to derive conclusions; *c) Review-based*: This method involves a systematic review and synthesis of existing literature on a topic. Additionally, the machine learning algorithms are categorized based on their taxonomies:

- **Supervised, Unsupervised, Semi-supervised, and Reinforcement Learning**: Foundational learning approaches where algorithms are trained using labeled data, unlabeled data, a mix of both, or by interacting with an environment. These methods have been applied to bolster both the infrastructure and application dimensions of blockchain (Jiang et al., 2016; Chen et al., 2021; Yang et al., 2022).

- **Causal Inference**: A method that discerns cause-and-effect relationships between variables, instrumental for policy evaluations within blockchain contexts (Liu et al., 2022; Zhang and Zhang, 2023).

- **Deep Learning Methods**: Neural network-based techniques with multiple layers designed for intricate pattern recognition, enhancing the efficiency of foundational learning paradigms. Within this category, *Large Language Models (LLMs)* are specialized deep learning models trained on extensive textual datasets, exhibiting proficiency in various natural language tasks. Their capabilities have been harnessed to augment blockchain security at the application layer (Gai et al., 2023; Gong, 2023).

- **Federated Learning**: A decentralized training approach where models are trained across multiple devices or servers without centralizing the data, offering enhancements across the

Table 1: Literature on Machine Learning for Blockchain

| Citation (year) | Facets | | | Methods | | | ML |
| --- | --- | --- | --- | --- | --- | --- | --- |
| | Infrastructure | Application | Cross-chain | Theoretical | Empirical | Review-based | Algorithm |
| Nadeem et al. 2023 | □ | ■ | □ | | | ✓ | explainable machine learning |
| Gai et al. 2023 | □ | ■ | □ | ✓ | ✓ | | large language model |
| Gong 2023 | □ | ■ | □ | ✓ | | | large language model |
| Liu and Zhang 2023 | □ | ■ | □ | ✓ | ✓ | | explainable machine learning |
| Zhu et al. 2023 | □ | ■ | □ | ✓ | ✓ | | federated learning |
| Zhang and Zhang 2023 | □ | ■ | □ | | | ✓ | causal inference |
| Liu et al. 2023 | □ | ■ | □ | ✓ | ✓ | | causal inference |
| Li et al. 2022 | □ | ■ | □ | ✓ | ✓ | ✓ | federated learning |
| Bar-Zur et al. 2022 | ■ | □ | □ | ✓ | ✓ | | reinforcement deep learning |
| Qu et al. 2022 | ■ | ■ | ■ | | | ✓ | federated learning |
| Yang et al. 2022 | □ | ■ | □ | | | ✓ | federated, reinforcement, supervised, unsupervised, semi-supervised learning |

[a] **Note:** The symbols in the table have specific meanings.
■: The topic is the primary focus of the paper.
□: The topic is mentioned or discussed but not the main focus.
✓: The paper covers the theoretical, empirical, or/and review-based methods. The theoretical method derives conclusions based on logical reasoning without relying on data. The empirical method uses data collection and analysis from observations or experiments to draw conclusions. The review-based method systematically examines existing literature to identify patterns or gaps in knowledge.

Table 2: Literature on Machine Learning for Blockchain (continued)

| Citation | Facets | | | Methods | | | ML |
|---|---|---|---|---|---|---|---|
| (year) | Infrastructure | Application | Cross-chain | Theoretical | Empirical | Review-based | Algorithm |
| Wang et al. 2021 | ■ | □ | □ | ✓ | ✓ | | multi-dimensional reinforcement learning |
| Chen et al. 2021 | ■ | ■ | □ | | | ✓ | supervised and unsupervised learning with or without deep methods, reinforcement learning |
| Mallouh et al. 2021 | ■ | □ | □ | ✓ | ✓ | | deep reinforcement learning |
| Liu et al. 2020 | □ | ■ | □ | | | ✓ | supervised, unsupervised, semi-supervised, reinforcement learning |
| Waheed et al. 2020 | □ | ■ | □ | | | ✓ | deep learning |
| Salah et al. 2019 | □ | ■ | □ | | | ✓ | AI and machine learning in general |
| Tanwar et al. 2019 | □ | ■ | □ | | | ✓ | unsupervised and deep learning |
| Dai et al. 2019 | □ | ■ | □ | ✓ | ✓ | | deep reinforcement learning |
| Hou et al. 2019 | ■ | □ | □ | ✓ | ✓ | | deep reinforcement learning |
| Jiang et al. 2016 | ■ | □ | □ | | | ✓ | supervised, unsupervised, semi-supervised, human-in-the-loop, game theory (GT)-based learning |

[a] **Note:** The symbols in the table have specific meanings.
■: The topic is the primary focus of the paper.
□: The topic is mentioned or discussed but not the main focus.
✓: The paper covers the theoretical, empirical, or/and review-based methods. The theoretical method derives conclusions based on logical reasoning without relying on data. The empirical method uses data collection and analysis from observations or experiments to draw conclusions. The review-based method systematically examines existing literature to identify patterns or gaps in knowledge.

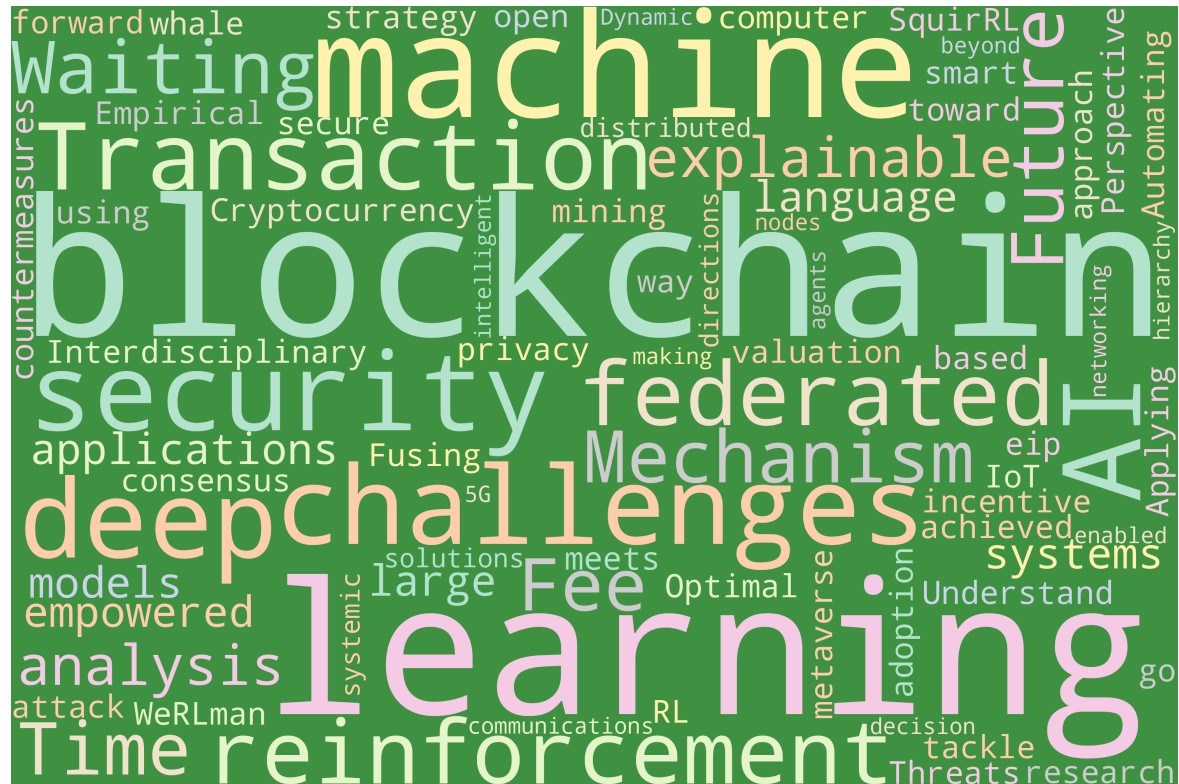

(a) The word cloud for the titles of literature in Table 1 and Table 2.

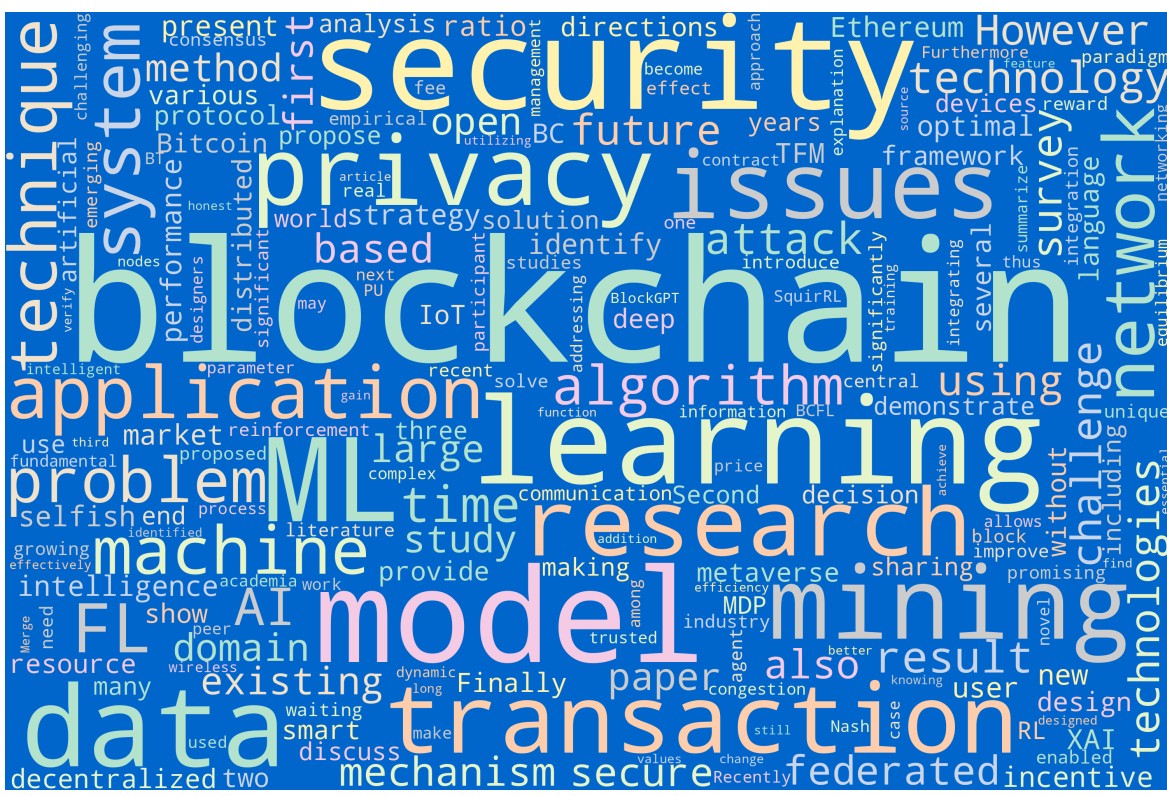

(b) Word cloud for the abstract of literature in Table 1 and Table 2.

**Note:** A word cloud is a visual representation of text data where the size of each word indicates its frequency or importance in the source text.

infrastructure, application, and cross-chain dimensions of blockchain (Qu et al., 2022; Yang et al., 2022; Li et al., 2022; Zhu et al., 2023).

- **Human-in-the-loop**: An approach that integrates human expertise into the AI learning process, particularly valuable for identifying and mitigating malicious threats to blockchain security (Jiang et al., 2016).
- **Strategic Thinking Dimensions**: Algorithms that incorporate strategic decision-making processes, are essential for fortifying blockchain security in scenarios where stakeholders engage in strategic interactions (Jiang et al., 2016).

Together, these categorizations provide a comprehensive overview of the current state of research in this domain. Figure 2a and Figure 2b further represent the word cloud of the title and abstract of the literature in Table 1 and Table 2. We have released the data and code for enhanced replicability on GitHub. You can access it at: `https://github.com/sunshineluyao/ml4blockchain`.

## 3 Future Research Directions

The integration of artificial intelligence (AI) with blockchain presents a myriad of opportunities to enhance the capabilities and applications of both domains. While there have been significant strides in this interdisciplinary area, several facets remain underexplored and warrant further investigation:

### 3.1 Causal Machine Learning

Despite the potential of causal inference in blockchain analysis, only a handful of studies have ventured into this domain (Liu et al., 2022; Zhang and Zhang, 2023), and many recent advancements in causal machine learning remain untouched. Traditionally employed in economics and social sciences to derive policy implications, causal inference can offer valuable insights when applied to blockchain's vast datasets. By leveraging causal machine learning (Kaddour et al., 2022; Schölkopf et al., 2021), it is conceivable to refine blockchain protocol designs and unearth superior solutions that cater to the unique challenges and requirements of decentralized systems.

### 3.2 Reinforcement Mechanism Design

Reinforcement learning has been sporadically applied to enhance the agents participating in the blockchain system (Bar-Zur et al., 2022; Yang et al., 2022; Hou et al., 2019; Dai et al., 2019; Mallouh et al., 2021; Wang et al., 2021). However, the focus has predominantly been on the agents rather than the blockchain protocol itself. There lies an untapped potential in equipping blockchain protocols with reinforcement learning capabilities, enabling them to autonomously search for and converge to optimal designs, a method called reinforcement mechanism design where the mechanism designer is a reinforcement learning agent (Tang, 2017; Cai et al., 2018). Such an approach could revolutionize the adaptability and efficiency of blockchain systems, ensuring they remain robust and relevant in dynamic environments (Zhang and Tian, 2022).

### 3.3 Cooperative AI

The notion of integrating human expertise into AI solutions on the blockchain has been touched upon in limited studies. Yet, as blockchain emerges as a form of decentralized AI, it inherently carries the ethical concerns and limitations associated with AI systems such as gender bias (Zhang et al., 2023b), crimes (Cong et al., 2023; Trozze et al., 2022), and other fairness issues (Zhang et al., 2022; Ao et al., 2022; Fu et al., 2023; Zhang et al., 2023a; Zhang, 2023). These concerns accentuate the need for human judgment in decision-making processes. Future research should pivot towards more human-in-the-loop solutions, ensuring that blockchain systems are ethical, transparent, and accountable. Additionally, as the web3 ecosystem grows and attracts a diverse array of stakeholders, there is a pressing need to incorporate strategic reasoning to cater to the sophisticated needs and expectations of these participants.

Parallelly, federated learning presents a compelling approach to enhancing blockchain's capabilities. By promoting collaborative learning across blockchain nodes, diverse blockchains, and even external systems, federated learning capitalizes on the collective intelligence of models Zhu et al. (2023);

Li et al. (2022); Yang et al. (2022). This method champions data privacy, with data remaining localized, obviating centralized storage and concurrently reducing storage overheads. Notably, much of the current research on federated learning and blockchain remains conceptual, with tangible implementations in public blockchains still in their infancy. Yet, as blockchain systems continue to diversify and grow, the synergy between federated learning's decentralized ethos and blockchain's foundational principles heralds a promising trajectory for future advancements.

In conclusion, the fusion of AI and blockchain offers a promising avenue for innovation and advancement. By addressing the aforementioned facets, future research can pave the way for more robust, efficient, and ethically sound blockchain systems.

## 4 Conclusion

This paper highlights the transformative potential of merging AI and blockchain, technologies that individually have reshaped our digital realm. As AI brings adaptability and learning, blockchain offers unparalleled security and decentralization. Their integration promises a digital infrastructure that's both adaptive and trustless. Despite blockchain's revolutionary promise, its rigid algorithms call for the dynamism of AI. We've delved deep into this convergence, emphasizing the need for AI-enhanced blockchain, particularly in smart contracts. Looking ahead, championing the synergy between these technologies will be pivotal for addressing modern challenges and crafting a future-ready, decentralized web3 ecosystem.

Conversely, the burgeoning blockchain economy yields vast quantities of open-source data, ripe for sophisticated machine learning algorithms, including deep learning and expansive language models. Future investigations should delve into how, in turn, blockchain might enhance AI capabilities.

## Acknowledgments and Disclosure of Funding

I extend my heartfelt gratitude to the anonymous reviewers of the NeurIPS 2023 AI for Science Workshop for their astute and invaluable comments. Their feedback has significantly contributed to the enhancement of this work. Furthermore, I am deeply appreciative of the stimulating intellectual exchanges that took place during the Electronic Information & Data Science Joint Workshop. This collaborative event, co-hosted by the Data Science Research Center at Duke Kunshan University and the School of Electronic Information at Wuhan University, convened at Duke Kunshan University on November 3rd, 2023, has enriched my perspectives and informed the research immensely. Luyao Zhang is supported by National Science Foundation China on the project entitled "Trust Mechanism Design on Blockchain: An Interdisciplinary Approach of Game Theory, Reinforcement Learning, and Human-AI Interactions (Grant No. 12201266) ."

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
