# OpenReview forum: "Machine Learning for Blockchain: Literature Review and Open Research Questions"
_NeurIPS.cc/2023/Workshop/AI4Science — NeurIPS2023-AI4Science Poster_

### Official Review · Reviewer_no9i · 2023-10-12
**Seems to be a very good, broad overview of the current literature of machine learning and blockchain**

**Rating:** 5
**Confidence:** 2

**Review:**

From my educated guess as a person who doesn't research nor have a lot of experience with blockchain, it seems to me that this paper offers a very good overview of the current literature in integrating both ML and the blockchain technologies, under the context of the web3 ecosystem. In this sense, all the figures, references, and tables seem to be very well created and thus this paper fits the objectives of the workshop.

Despite what seems to be a very well curated and explored analysis of the current literature and state of the art, the paper's abstract and introduction seem to defend that the utility of this paper is, instead, to justify why it is so important to fuse ML and the blockchain technologies. I don't see how this paper justifies this need in any way, nor why ML is such an important tool to solve these needs. In other words, there's clearly research going on ML applied to blockchain, but why is this technology (or, as the authors call it, "enhanced intelligence") so important and distinct for blockchain? Isn't there other approaches to improve infrastructure, security, efficiency, and interoperability? If the authors do believe that this is the case, then I'd argue they should have defended why that's the case, instead of showing that there are gaps in the ML/blockchain literature, and somehow that is enough to demonstrate that integrating ML into blockchain is such a dire need.

From my limited knowledge of blockchain, the usual complains I hear from my peers is that the technology has a huge problem  of scalability and speed, but this paper doesn't seem to mention these problems at all. Aren't these two problems much more important to blockchain than the integration of "enhanced intelligence"?

Another issue that I think it's a much more pressing problem is what is said in line 26, in which it is said that AI brought personalised content recommendations and targeted advertising. This is presented in a way (from my understanding) as a very positive thing without a negative side. However, even if you argue that this was positive (to whom?), these personalised contents usually come at the cost of losing privacy, and a lot of personal data being collected without the full understanding of the users. I find this to be a very important, and current issue of AI, but the authors do not seem to mention this in this work and instead use this fact as a positive side of AI that somehow justifies it being brought to blockchain?

Finally, at the end of the introduction, the paper seems to defend that this paper explores ML's empowering role in the "infrastructure, application, and cross-chain dimensions, bolstering security, efficiency, and interoperability". This is a very strong statement and, again, I don't think this was fully supported in the paper. Beyond what I've already mentioned that the paper seems to show possible research directions in these topics, but not exactly why ML should be *the* method to fully solve these issues instead of other methods. I think this link is even weaker for the cases of security, efficiency, and interoperability, as they are briefly mentioned only in the context of the main topics defined in section 2 and 3.

All in all, for me this paper seems to fail for the usual hyped theme that AI/ML will solve all the problems in the world, without looking to (what I think) are more pressing issues. Likewise, it goes into the hype without actually trying to show why this approach is better than any other, or whether this is the only approach possible in the context of blockchain (from what I hear from my colleagues working with blockchain, doesn't seem like AI and ML are the only tools available).

If this work was framed as a literature revision of ML and blockchain, and what the open research directions are, I think this would have been a very good paper; however, the fact that it is framed as a "justification" of AI/ML as amazing ways to solve blockchain current issues, makes this a paper that I cannot recommend for acceptance.

---

### Official Review · Reviewer_NCry · 2023-10-23
**Talks about different ways in which machine learning can improve blockchain technologies**

**Rating:** 7
**Confidence:** 4

**Review:**

Pros

Well structured and easy to follow through.
The table illustrating the literature review is very useful.

Cons

There's an overlap in the listing of machine learning methods.  For instance, deep learning methods could be listed under supervised, unsupervised, semi-supervised, etc

---

### Meta-Review · Area_Chair_Kpef · 2023-10-26

**Recommendation:** Accept (Poster)
**Confidence:** 3

**Metareview:**

The paper prompts the idea that AI/ML can solve all problems, without adequately addressing more critical issues. This paper fails to demonstrate the superiority of the AI/ML approach in the context of blockchain, suggesting that other tools may be equally valid.

The paper has been accepted for the workshop because its content is deemed suitable for the audience, particularly those without prior blockchain experience.